# **Development of CarbonTracker Europe-CH**<sub>4</sub> – Part 1: system set-up and sensitivity analyses

Aki Tsuruta<sup>1</sup>, Tuula Aalto<sup>1</sup>, Leif Backman<sup>1</sup>, Janne Hakkarainen<sup>2</sup>, Ingrid T. van der Laan-Luijkx<sup>3</sup>, Maarten C. Krol<sup>3,5,6</sup>, Renato Spahni<sup>4</sup>, Sander Houweling<sup>5,6</sup>, Marko Laine<sup>2</sup>, Marcel van der Schoot<sup>7</sup>, Ray Langenfelds<sup>7</sup>, Raymond Ellul<sup>8</sup>, Wouter Peters<sup>3,9</sup>

5

<sup>1</sup>Climate Research, Finnish Meteorological Institute, Helsinki, Finland <sup>2</sup>Earth Observation, Finnish Meteorological Institute, Helsinki, Finland <sup>3</sup>Meteorology and Air Quality, Wageningen University, Wageningen, the Netherlands <sup>4</sup>Climate and Environmental Physics, Physics Institute, and Oeschger Centre for Climate Change Research, University of 10 Bern, Bern, Switzerland <sup>5</sup>SRON Netherlands Institute for Space Research, Utrecht, the Netherlands <sup>6</sup>Institute for Marine and Atmospheric Research, Utrecht University, Utrecht, the Netherlands <sup>7</sup>CSIRO Oceans and Atmosphere, Aspendale, Australia <sup>8</sup>Atmospheric Research, Department of Geosciences, University of Malta, Msida, Malta

<sup>9</sup>Centre for Isotope Research, University of Groningen, Groningen, the Netherlands 15

Correspondence to: Aki Tsuruta (Aki.Tsuruta@fmi.fi)

Abstract. CarbonTracker Europe-CH<sub>4</sub> (CTE-CH<sub>4</sub>) inverse model versions 1.0 and 1.1 are presented. The model optimizes global surface methane emissions from biosphere and anthropogenic sources using an ensemble Kalman filter (EnKF) based optimization method, using the TM5 chemistry transport model as an observation operator, and assimilating global in-situ

- atmospheric methane mole fraction observations. In this study, we examine sensitivity of our CH<sub>4</sub> emission estimates on the 20 ensemble size, covariance matrix, prior estimates, observations to be assimilated, assimilation window length, convection scheme in TM5, and model structure in the emission estimates by performing CTE-CH4 with several set-ups. The analyses show that the model is sensitive to most of the parameters and inputs that were examined. Firstly, using a large enough ensemble size stabilises the results. Secondly, using an informative covariance matrix reduces uncertainty estimates. Thirdly,
- agreement with discrete observations became better when assimilating continuous observations. Finally, the posterior 25 emissions were found sensitive to the choice of prior estimates, convection scheme and model structure, particularly to their spatial distribution. The distribution of posterior mole fractions derived from posterior emissions is consistent with the observations to the extent prescribed in the various covariance estimates, indicating a satisfactory performance of our system.

#### **1** Introduction 30

Inverse modelling is a popular tool to estimate global to regional scale surface greenhouse gas fluxes. The models are often based on assimilation techniques, where surface emissions are constrained by measurements of atmospheric mole fractions.

The inversion method can provide information on the emissions, consistent with measurements of atmospheric mole fractions. This allows greenhouse gas emissions to be studied in areas where local measurements are absent, and environmental drivers of emissions are unknown. Furthermore, greenhouse gas emission estimates are uncertain because not all processes related to gas emissions and uptakes are understood. Using inversion methods, we are generally able to reduce uncertainties in existing estimates.

- Several inverse models have recently been developed to estimate global methane (CH<sub>4</sub>) surface emissions (e.g. Bergamaschi *et al.* (2009), LMDZ-MIOP (Bousquet *et al.*, 2006) LMDZt-SACS (Pison *et al.*, 2009; Bousquet *et al.*, 2011), CarbonTracker-CH<sub>4</sub> (Bruhwiler *et al.*, 2014; Tsuruta *et al.*, 2015), GEOS-Chem (Fraser *et al.*, 2013), MATCH model (Chen
- and Prinn, 2006)). Those models differ in many ways, such as optimization technique (e.g. an adjoint model 4DVAR, ensemble Kalman smoother/filter), transport model (e.g. TM5, LMDZ, GEOS-Chem, MATCH) that is used as an observation operator, prior emission estimates, and set of observations assimilated to constrain the emissions. Although the inverse modelling technique has the advantage that the estimates are consistent with the observations, the emission estimates can vary between the models (Kirschke *et al.*, 2013). Therefore, independent estimates would be useful to further understand
- the trends and spatial patterns of methane emissions and its atmospheric concentrations, and relations between the two. Furthermore, the estimates can vary not only between the models, but also depending on the model settings.

Ideally, different inverse models should be able to produce the same emission estimates regardless of prior estimates. Bergamaschi *et al.* (2010, 2013) showed sensitivity of the derived emission estimates, but found that it was not significant
for regional and global estimates. However, at regional scale, spatial distribution is dependent on the prior estimates. Therefore, it is important to include as much information as possible about the spatial distributions in the prior emission estimates for those models that optimizes emissions region-wise (Bruhwiler *et al.*, 2014; Tsuruta *et al.*, 2015). Because of the nature of inverse models that are constrained by the observations, it is very important to use observations that cover the domain of interest in time and space well. Previous studies showed that increasing the number of methane observations
improved the emission estimates (Beck *et al.*, 2012; Bergamaschi *et al.*, 2013; Houweling *et al.*, 2014; Fraser *et al.*, 2013).

- Furthermore, Locatelli *et al.* (2013) addressed the impacts of the transport model by comparing flux estimates made with the same system, but with different transport models. Although the estimates agreed well for the global total, regional estimates differed by 23-48%. All of these impacts may vary between the models, and the total differences are more than the sum of each part, and an impact in a certain set-up may not be quantified by estimating it in another set-up. Therefore, it is important
- 30 to assess those impacts when setting up a new model, such as our attempt here.

In this study, we present impact of parameters and inputs on methane emission estimates using the CarbonTracker Europe-CH<sub>4</sub> (CTE-CH<sub>4</sub>) model. CTE-CH<sub>4</sub> is a methane version of CarbonTracker (Peters *et al.* 2005), originally developed to optimize global CO<sub>2</sub> fluxes by NOAA-ESRL (National Oceanic and Atmospheric Administration's Earth System Research

Laboratory) and Wageningen University, the Netherlands. CarbonTracker was later implemented for CH<sub>4</sub> by the NOAA-ESRL (Bruhwiler et al., 2014) and the Finnish Meteorological Institute (FMI) (Tsuruta et al., 2015). CTE-CH<sub>4</sub> is based on an ensemble Kalman filter (EnKF; Evensen, 2003) optimization technique and uses the TM5 atmospheric chemistry transport model as an observation operator. EnKF is a widely applied method in inverse models, when the dimension of the state vector and hence its covariance matrix is large. TM5 is a well-established atmospheric transport model that is used both as a forward model and an observation operator in many other systems, and is under active development. CTE-CH<sub>4</sub> optimises anthropogenic and biosphere methane emission estimates, constrained by global in-situ atmospheric CH<sub>4</sub> mole fraction observations. CTE-CH<sub>4</sub> is built to work well globally, but has its focus on Europe. The zoom of TM5 is applied over Europe, where the observation network is dense. Furthermore, Europe is divided into four sub-regions in which the fluxes are optimised separately.

As CTE-CH<sub>4</sub> is an EnKF based system, we assessed the impacts of ensemble size and state vector covariance matrix using several ensemble sizes and two different covariance matrices. Furthermore, in order to assess the impacts of prior emission estimates, observations, transport model, and the model structure, we performed CTE-CH<sub>4</sub> using two sets of biosphere prior

- emission estimates, two sets of observations, two convection schemes in TM5, and two versions of the CTE-CH4 model. To 15 limit the computational cost, summer 2007 was selected as a study period. The aim of this study is to introduce the set-up and choices made for an optimally working methane inversion system, which will later be used in long term studies of global and regional methane emissions and their trends presented in an accompanying paper (Tsuruta et al., 2016). The results and evaluation of longer time series (recent decade), comparison between two model structures, and detailed analysis on estimates over Europe will be discussed in the second part of this paper. 20

25

30

5

10

# 2. Methods and datasets

#### 2.1 CTE-CH<sub>4</sub> data assimilation system

CTE-CH<sub>4</sub> is a data assimilation system that optimizes global surface methane emissions by minimizing a cost function:

$$J = (\mathbf{x} - \mathbf{x}^{\mathbf{b}})^{T} \mathbf{P}^{-1} (\mathbf{x} - \mathbf{x}^{\mathbf{b}}) + (\mathbf{y} - H(\mathbf{x}))^{T} \mathbf{R}^{-1} (\mathbf{y} - H(\mathbf{x}))$$
(1)  
$$\mathbf{E} = G(\mathbf{x}) \mathbf{E}^{\mathbf{b}}$$
(2)

where  $\mathbf{x}$  (dimension N) is a state vector that contains a set of scaling factors that multiply the CH<sub>4</sub> surface emissions ( $\mathbf{E}$ , dimension  $360 \times 180$ ) that we want to improve starting from a prior estimate of these emissions ( $E^{b}$  [ $360 \times 180$ ]) and scaling factors  $x^{b}$  [N]. **P** [N×N] is the covariance matrix of the state vector, **y** (dimension M) is a vector of atmospheric methane mole fraction observations,  $\boldsymbol{R}$  [M×M] is a covariance matrix of the observations  $\boldsymbol{y}$ , and  $\boldsymbol{H}$  is an observation operator [M×N]. The operator G transforms the regionally estimated scaling factors x to a  $1^{\circ} \times 1^{\circ}$  global map, which are used to scale prior emissions E. The cost function in Eq. (1) is minimized using an ensemble Kalman filter (EnKF) (Evensen, 2003) with 500

ensemble members, and the TM5 chemistry transport model (Krol *et al.*, 2005) was used as an observation operator that transforms emissions E into simulated CH<sub>4</sub> mole fractions (H(x)). The emissions E were optimized weekly, with an assimilation window length of 5 weeks.

The optimal weekly mean CH<sub>4</sub> fluxes  $F_{tot}$  at region r, and time (week) t were calculated as follows:

 $F_{tot}(r,t) = \lambda_{bio}(r,t) \times F_{bio}(r,t) + \lambda_{anth}(r,t) \times F_{anth}(r,t) + F_{fire}(r,t) + F_{term}(r,t) + F_{oce}(r,t)$ (1)

where  $F_{bio}$ ,  $F_{ant}$ ,  $F_{fire}$ ,  $F_{term}$ ,  $F_{oce}$ , are the prior emissions from biosphere, anthropogenic activities, fire, termites and ocean, respectively.

- The regional definition of CTE-CH<sub>4</sub> is defined based on modified TransCom (mTC) and land-ecosystem regions (Fig. 1). The mTC regions were defined similarly to Tsuruta *et al.* (2015). Land-ecosystem regions in 1°×1° grid were defined based on Prigent *et al.* (2007) and Wania *et al.* (2010), as in the LPJ-WHyME vegetation model (Spahni *et al.* 2011), and contains six land ecosystem types (LET): inundated wetland and peatland (IWP), wet mineral soil (WMS), rice (RIC), anthropogenic land (ANT), water (WTR) and ice (ICE). WTR is defined similarly to Peters *et al.* (2007), and the ICE region corresponds to
- the ice region in the mTC definition. The rest of land-ecosystem regions were defined according to the fraction of IWP, WMS and RIC used in LPJ-WHyME. To limit the number of degrees of freedom, only one dominant LET was assigned to each grid cell. In the following cases, LET with the largest fraction was chosen. For grid cells where fraction of IWP, WMS or RIC is larger than 0.1, either IWP, WMS or RIC was assigned. For grid cells where the fraction of IWP or WMS are smaller than 0.1, and the prior anthropogenic emission estimates (EDGARv4.2 FT2010, see Section 2.3) including emissions
- from rice fields are zero, IWP or WMS was assigned. Furthermore, if the LPJ-WHyME biosphere emission estimates are much larger than the EDGARv4.2 FT2010 emission estimates (more than 200%), either IWP or WMS was assigned. On the other hand, if the EDGARv4.2 FT2010 emission estimates were much larger than LPJ-WHyME biosphere emission estimates, either ANT, RIC or WTR were assigned.
- For the baseline model (M1), biosphere emissions were optimized in regions where LET are either IWP or WMS (i.e.  $\lambda_{anth}(r,t) = 0$ ), and anthropogenic emissions were optimized in regions where LET are RIC, ANT or WTR (i.e.  $\lambda_{bio}(r,t) = 0$ ). This approach resulted in 28 biosphere regions and 30 anthropogenic regions, i.e. 58 scaling factors  $\lambda(t) = (\lambda_{bio}(t), \lambda_{anth}(t))$  to be optimized per week globally. This number of scaling factors is smaller than theoretically expected (16 mTC regions × 5 land-ecosystem regions = 80 scaling factors) because some mTC regions contain less than five
- ecosystems types. For comparison, we developed a model where both  $\lambda_{bio}(r, t)$  and  $\lambda_{anth}(r, t)$  are optimized in each region (M2). In M2, the regional definition of the scaling factors for the biosphere emissions is based on the combination of mTC and land-ecosystem regions (58 regions), and for the anthropogenic emissions, the mTC region (15 regions) is used. This results in 74 scaling factors to be optimized per week globally in M2. Note that scaling factors are estimated purely

mathematically in EnFK, and there is no system to choose which scaling factors  $(\lambda_{bio}(r,t) \text{ or } \lambda_{anth}(r,t))$  gain larger weights based on physical or meteorological properties.

# 2.2 TM5 atmospheric chemistry transport model

- The link between atmospheric CH<sub>4</sub> observations and the exchange of CH<sub>4</sub> at Earth's surface is transport in the atmosphere. In
  our data assimilation system, the TM5 chemistry transport model version 3.0 (Krol *et al.*, 2005; Huijnen *et al.*, 2010) was applied as the observation operator. TM5 was run with a 1°×1° (latitude x longitude) zoom region over Europe (24°N to 74°N, 21°W to 45°E), framed by an intermediate zoom region of 2°×3°, and a global 4°×6° degree resolution, driven by 3-hourly ECMWF ERA-Interim meteorological fields with 25 vertical layers. The main sink of CH<sub>4</sub> atmospheric chemical methane loss by OH in the troposphere, was calculated using off-line chemistry with monthly tropospheric OH concentrations concentrations based on Spivakovsky (2000) scaled by 0.92 (Houweling *et al.*, 2014), based on the inversion study by Huijnen *et al.* (2010). Furthermore, stratospheric methane sinks due to reaction with OH, Cl and O(<sup>1</sup>D) were included by applying reaction rates based on the 2D photochemical Max-Planck-Institute (MPI) model (Brühl and Crutzen, 1993). The global total atmospheric chemical loss, i.e. the integrated OH, Cl and O(<sup>1</sup>D) losses during the test period was 229 Tg CH<sub>4</sub> (per 154 days).

In order to avoid influence of initial condition of atmospheric concentrations, TM5 was run as a forward mode for model year of 2007, starting from uniform mole fraction of 1600 ppb globally and using prior emission estimates. Using its final values, i.e. assigning the values of the end of the model year as an initial condition, TM5 was run once again in a forward mode for 2007. Using the second final values, CTE-CH<sub>4</sub> was run starting from the beginning of 2007 up to June 2007 using set-up S1 (see Section 3.2) to get well-mixed initial atmospheric concentrations for the experiments presented in this study.

TM5 was recently equipped with two versions of convection schemes. A version based on Tiedtke (1989), and another version that takes the convection mass fluxes directly from ECMWF ERA-Interim (Gregory *et al.*, 2000). The latter approach has a stronger convection in the northern hemisphere during summer, producing lower concentrations near the

surface, and higher concentrations in the free troposphere and lower stratosphere compared to the former version (Olivie *et al.*, 2004).

#### 2.3 Prior CH<sub>4</sub> flux datasets

The prior methane emission estimates  $F_{anth}$ ,  $F_{bio}$ ,  $F_{fire}$  and  $F_{term}$  in Eq. (1) were collected from inventories and previous studies. All emission fields were processed to match the finest TM5 resolution, i.e.  $1^{\circ} \times 1^{\circ}$  (latitude × longitude), globally. The total prior emissions for each category are given in Table 3.

Anthropogenic methane emissions are responsible for more than half of the global methane source. For monthly mean anthropogenic emissions, the Emission Database for Global Atmospheric Research version 4.2 (EDGARv4.2) FT2010 is used. The emissions from agricultural waste burning and large scale biomass-burning were removed, because it overlaps with fire emissions described later. The latest version of the EDGAR inventory contains the longest time series, which is an advantage for our long-term simulation described in a follow up study (Tsuruta *et al.*, 2016). To be consistent, we applied

5 advantage for our long-term simulation described in a follow up study (Tsuruta *et al.*, 2016). To be consistent, we applied this version of EDGAR inventory data also in this study. We did not introduce monthly variations in the yearly prior anthropogenic emissions.

Biosphere emissions, dominated by wetlands, are estimated to contribute about 40% of the total CH<sub>4</sub> emissions, where interannual variability of emissions from wetland ecosystems is estimated to be 12 Tg CH<sub>4</sub> yr<sup>-1</sup> (Spahni *et al.*, 2011). For the monthly mean biosphere emissions, the estimates from two biogeochemical process models, LPX-Bern 1.0 (Spahni *et al.*, 2013) and the LPJ-WHyME vegetation model (Spahni *et al.*, 2011), were used to test the impacts of the prior emissions on the results. The emissions from rice fields were removed from both estimates since they were already included in the anthropogenic emissions. EDGARv4.2 FT2010 estimates of the emissions from rice fields were about 13 Tg CH<sub>4</sub> yr<sup>-1</sup>
smaller than that of LPJ-WHyME and LPX-Bern estimates; no scaling was applied to the rice field emissions from the

EDGARv4.2 FT2010 estimates.

Methane emission from fire, termites and ocean accounts for only about 20% of the global estimates in total (Kirschke *et al.*, 2013). However, their temporal and spatial variability are large and should be accounted for in the model. Inter-annual variability of emission from fire can be large because of occasional intense fire years, and events such as strong El Niño conditions that lead to dry periods around the Equator (Langenfelds *et al.*, 2002). Also, seasonal and spatial patterns of fire vary due to e.g. vegetation types, changing synoptic weather patterns, agricultural practices, and deforestation (van der Werf *et al.*, 2010). The habitat of termites depends on vegetation types, and the amount of methane emitted from termites depends on the species (Eggleton and Tayasu, 2001).

25

For fire emissions, monthly mean estimates from the Global Fire Emissions Database version 3.1 (GFEDv3.1) (van der Werf *et al.*, 2010) were used. For termites emission, annual mean estimates from Ito *et al.* (2012) were used. For ocean emissions, monthly mean estimates were calculated as the product of the difference between air and seawater partial pressures of methane (Lambert and Schmidt, 1993), gas transfer velocity, gas solubility. ECMWF ERA-Interim sea surface temperature,

30 sea ice mask, surface pressure and wind speed (Dee *et al.*, 2011) was used for calculating solubility and transfer velocity. The saturation of methane in the surface seawater was assumed to be constant globally and throughout the months.

#### 2.4 Atmospheric methane observations

Atmospheric observations of CH<sub>4</sub> dry-air mole fractions, gathered by a large community of dedicated experimentalists worldwide and made available from the World Data Centre for Greenhouse Gases (WDCGG), were assimilated in CTE-CH<sub>4</sub>. The set of observations consists of discrete air sample and continuous measurements from several cooperative networks (Table 1). Observations capturing 'background' signatures of atmospheric CH<sub>4</sub> were selected, and CH<sub>4</sub> values suspected to have strong local influences were excluded using information provided by the experimentalists who conducted the original quality control on their data. For continuous observations, daytime (12-16 local time) observations were selected, except for the high altitude sites (PRS, CMN, SCH, PUY, IZO, JFJ, MLO, ZUG, ZSF) for which night time (0-4 local time) observations were selected. These choices of sampling times reflect a preference for well-mixed conditions that represent

- large source areas, and are also better captured by the TM5 transport model. No day-night selection was applied to discrete observations. For each site, model-data-mismatches (mdm), used in the observation covariance matrix, were defined based on uncertainty in the observations and the transport model and are given in Table 1. This includes both the observation error and the transport model error, i.e. ability of the transport model to simulate the observations. Note that the latter error is often much larger than the former. For the marine boundary layer and the deep southern hemispheric sites, mdm was set 7.5 ppb,
- for the sites which captures both land and ocean signals, mdm was set 15 ppb, for the sites capturing signals from land, mdm was set 25 ppb, for the sites with large variation in observations due to local influences, mdm was set 30 ppb, and for the sites appeared problematic in the inversions, mdm was set 75 ppb. During assimilation, rejection thresholds were set as three times mdm.

# 3. Experiments set-ups

In this study, inversions were performed for a test period between 29 May 2007 and 30 Oct. 2007. Summer was chosen because it is the time when the biospheric  $CH_4$  emissions are largest in the northern hemisphere, and our focus is on the northern boreal region and Europe.

#### 3.1 EnKF parameters' sensitivity experiments

Two EnKF parameters (ensemble size and prior covariance matrix) were assessed using the baseline model, M1, with only discrete air sample observations assimilated, and prior biosphere emission estimates from the LPX-Bern (S3). EnKF has a property that a full posterior probability density function of the state (scaling factor in our case) can be exactly represented by an infinite ensemble of model states. However, computational constraints limit the number of ensemble members. A small number of ensemble members is computationally cheap to apply, but it may lead to a statistical misrepresentation of the posterior distribution. Choosing the suitable number of ensembles is often a question of finding a large enough number of

30 ensembles that is not computationally too expensive. For the sensitivity experiments, we used an ensemble size of 20 (E20) and 500 (E500), and additionally made a specific test for degrees of freedom related to five different ensemble sizes from 20

to 500 (20, 60, 120, 240, 500). FMI has a computer facility with 20 nodes per processor. For E20, one processor is used, and for E500, 13 processors were used. In order to test sensitivities of the prior distribution of the states, we carried out four E20 simulations and three E500 simulations using random initial values sampled from normal distribution with mean 1 and standard deviation 1.

5

A model error covariance matrix Q is used to create a prior state covariance matrix at the beginning of each time step:

$$\boldsymbol{P}_{b}^{t+1} = \boldsymbol{P}_{a}^{t} + \boldsymbol{Q},\tag{2}$$

where  $P_b^{t+1}$  is the prior state covariance matrix at time t + 1, and  $P_a^t$  is posterior state covariance matrix at time t. Two matrices were examined in this study: identity (Q1), and Q2 which was based on Peters *et al.* (2005):

$$Q2 = \begin{pmatrix} A_{IWP} & A^{*1} & 0 & 0 & 0 \\ A^{*1} & A_{WMS} & 0 & 0 & 0 \\ 0 & 0 & A_{ANT} & A^{*2} & 0 \\ 0 & 0 & A^{*2} & A_{RIC} & 0 \\ 0 & 0 & 0 & \sigma_{ICE} \end{pmatrix},$$
$$A_{k_{ij}} = \begin{pmatrix} \sigma_k^2 & \sigma_k^2 \cdot e^{-d_{ij}/L} \\ \sigma_k^2 \cdot e^{-d_{ij}/L} & \sigma_k^2 \end{pmatrix} \text{ for } k = \text{IWP, WMS, ANT, RIC.}$$

It is assumed that  $\lambda_{IWP}$ ,  $\lambda_{WMS}$ ,  $\lambda_{ANT}$ ,  $\lambda_{RIC}$ ,  $\lambda_{ICE}$  are uncorrelated, with each having a variance  $\sigma_k^2 = 0.8$ . Scaling factors of the same LET regions at different mTC regions (off diagonal of  $A_{k_{ij}}$ ) are assumed to be correlated with  $\sigma_k \cdot e^{-d_{ij}/L}$ , where  $d_{ij}$  is the distance between the centre of the regions (*i*, *j*), and the correlation length L = 900km. For mTC3 (south American

tropical), 7 (Eurasian boreal), and mTC9 (Asian tropical), between  $\lambda_{IWP}$  and  $\lambda_{WMS}$  ( $A^{*1}$ ), and between  $\lambda_{ANT}$  and  $\lambda_{RIC}$  ( $A^{*2}$ ) are assumed correlated with  $\sigma_k^2 \cdot e^{-d_{ij}/L}$  to constrain the emissions in those regions better. The observation network within and around these regions are particularly sparse (only one or no site in the regions), which makes the model difficult to constrain the emissions. For  $\lambda_{ICE}$ , variance  $\sigma_{ICE}^2$  is set to be  $1e^{-8}$  for both **Q1** and **Q2**, as the emissions from this region are small, and we assume the prior estimates are already good.

# 20 3.2 Other sensitivity experiments

In the following experiments, an ensemble size of 500, the same set of prior state distribution sampled from the same normal distribution with mean 1 and standard deviation 1 (i.e. no random error due to sampling of prior state), and Q2 covariance were used. In the reference inversion S1, the emissions were estimated using model M1 with an assimilation window length of 5 weeks (i.e. taking into account the observations performed within the corresponding week and the next

25 four ones), with both discrete and continuous methane observations assimilated, using the LPX-Bern emission estimates as the prior biosphere emissions, and the Tiedtke (1989) convection scheme in the TM5 transport model. For sensitivity analysis, seven inversions were performed (Table 2). S2 examines the effects of prior biosphere emissions by replacing the LPX-Bern emissions with the LPJ-WHyME emission estimates. S3 examines the effects of using continuous observations by

assimilating only discrete observations, and S4 examines effects of the assimilation window length by increasing it to 12 weeks instead of 5 weeks. Additionally, the inversions using the M2 model examine effects of increasing the number of scaling factors and simultaneously optimising two scaling factors per region in the model structure (S5). Finally, S6 and S7 examines the effect of the updated convection scheme in TM5 using M1 and M2, respectively.

# 5 4. Results

# 4.2 Sensitivity of EnKF parameters

The results from sensitivity runs (E20-E500) show that the larger the ensemble size, the more stable the results are. With an ensemble size of 500, the mean estimates for the sum of the biosphere and anthropogenic emissions aggregated over the test period only differed by less than 0.5 Tg CH<sub>4</sub> between the three E500 runs (217.9 ± 28.2, 217.7 ± 28.2, 217.4 ± 27.3 Tg CH<sub>4</sub> 10 per test period). However, with ensemble size 20, the mean estimates for the aggregated sum of the biosphere and anthropogenic emissions differed by about 10 Tg CH<sub>4</sub> (216.7 ± 25.3, 221.0 ± 24.9, 224.4 ± 24.3, 225.1 ± 24.6 Tg CH<sub>4</sub>). The smaller posterior uncertainties in the E20 experiments than in the E500 experiments are the underestimation of the uncertainties due to the small ensemble size. The weekly sums also show that there are more random variations in the estimates from the E20 experiments compared to the E500 experiments (Fig. 2). These differences depend on the available observations. Regions with dense observational networks, e.g. north American boreal, show less variation in the estimates than where the observation network is sparse, e.g. Asian tropical. This holds for both E20 and E500. The degree of freedom (d.o.f.) in the posterior ensembles (square of sum of singular values divided by sum of square of singular values) is small

- when the ensemble size is small as we cannot represent more d.o.f. than we have in the ensemble members. It increases significantly until an ensemble size of 120, meaning the information added to the singular value decomposition matrix is
  significant, but the rate of increase slows down after that, reaching to an equilibrium (Fig. 3). Although we did not test larger ensemble sizes, the results suggest that 500 is large enough to represent the probability distribution well, and will be used in the follow-up experiments.
- The computational costs were higher for E500, as expected. With 13 processors of our computational system at FMI, the computational burden was about one hour of wall clock time per week of model time for E500. For E20, the burden was only about half an hour per week of model time with one processor. Note that the computational time of E500 could be as small as E20 if number of nodes is increased to 500, i.e. using 25 processors in case of FMI system. The observation operator was the most expensive, consuming about 80% of computational time for both cases.
- 30 The experiments using Q1 and Q2 prior covariance showed that the mean posterior emissions did not differ very much. The posterior emissions using Q1 were  $91 \pm 14$  Tg CH<sub>4</sub> for biosphere emissions and  $126 \pm 27$  Tg CH<sub>4</sub> for anthropogenic emissions (the numbers are aggregated over the entire run of 154 days). The estimates using Q2 were  $91 \pm 13$  Tg CH<sub>4</sub> for the