# Peer review of "Development of CarbonTracker Europe-CH4 – Part 1: system set-up and sensitivity analyses"

_Geoscientific Model Development, 2016_

## Short Comment (SC1) · 22 Aug 2016

Dear authors,

In my role as Executive editor of GMD, I would like to bring to your attention our Editorial version 1.1:

http://www.geosci-model-dev.net/8/3487/2015/gmd-8-3487-2015.html

This highlights some requirements of papers published in GMD, which is also available on the GMD website in the 'Manuscript Types' section:

http://www.geoscientific-model-development.net/submission/manuscript_types.html

In particular, please note that for your paper, the following requirement has not been met in the Discussions paper:

- "The main paper must give the model name and version number (or other unique identifier) in the title."

For a model evaluation it is important to know, which model version exactly was evaluated. Therefore, please add a version number for your model in the title upon your revised submission to GMD.

Yours,

Astrid Kerkweg

―――――――――――――――――――――

---

## Referee Comment (RC1) · Anonymous Referee #1 · 17 Sep 2016

The manuscript "Development of CarbonTracker Europe-CH4 – Part 1: system set-up and sensitivity analyses" by A. Tsuruta et al. presents a sensitivity study of the resulting methane fluxes to the underlying assumptions of the atmospheric inversion system. The authors analyse seven different inversion system set ups (by varying EnKF ensemble size, length of assimilation window, prior flux fields, prior error covariance matrix, structure of the control vector and convection scheme in the transport model) with respect to the estimated fluxes and their posterior uncertainties over a period of five months (June to October 2007).

Unfortunately, the manuscript is not well written and would improve from English language editing. There are also inconsistencies and misunderstandings especially in the

Introduction section; for example page 1, lines 31/32: which models are based on assimilation techniques? It is rather that in atmospheric transport inversions assimilation techniques are often employed to invert the transport. Another example just follows on page 2, line 1: the inversion method does not provide information on the emissions; it is rather the observations that provide information. I suggest the authors (some of the co-authors are native or close-to-native speakers and some are world-leading experts in inverse modelling) carefully go through the manuscript to improve the language and correct these misunderstandings of how transport inversions work.

The main problem of this manuscript, however, is its aim, as the authors state: 'The aim is to introduce the set-up and choices for an inversion system, which will be used in long term studies and presented in an accompanying paper.' Is this really sufficient for a publication in its own or could that have been merged into the accompanying paper that looks at long-term methane emissions and trends. And how much do we really learn from a sensitivity study covering a very short time period (5 months) with no interannual nor seasonal variability for an inversion aiming to analyse long-term emission fields and trends? On top of that the CTE-CH4 system has essentially already been published elsewhere (see Tsuruta et al., 2015).

Here are a few suggestions to improve the manuscript:

- Why did the authors finally choose S1 and S5 for the long-term experiment in the accompanying paper? Based on the evaluation in this manuscript here there are not any indications to rule out any of the tested configurations. At least the authors do not provide any objective reasons.

- Is there anything the inverse modelling community can learn from the experiments here in general? Maybe the results of some of the experiments can be analysed in more depth such that the findings can be generalised?

- It would be illustrative to run some longer experiments to account for seasonal and interannual variability for some selected set-ups.

- Although the title suggests that the inversion system focuses on Europe the analysis of the results in the manuscript does not. What is the effect of the zoom over Europe? And although you zoom over Europe you only distinguish between four regions.

- In Fig 4 plotting the relative differences in the uncertainty estimates would be much more illustrative.

- In Fig 5 the fit against the observations when transporting the posterior fluxes is not very impressive in this kind of plot. Maybe time series for some selected stations would illustrate the improvement much better.

---

## Author Comment (AC1) · 13 Dec 2016

In the following, referee's comments are in *italic*, authors' responses in normal font, and references (page, line, figure, and table number) to the revised manuscript in **bold**. Please note that this paper was merged with the accompanying paper, following the referees' comments and with approval from the Topical Editor. The summary of this paper was included in the Supplementary Material of the accompanying paper.

*Unfortunately, the manuscript is not well written and would improve from English language editing. There are also inconsistencies and misunderstandings especially in the Introduction section; for example page 1, lines 31/32: which models are based on assimilation techniques? It is rather that in atmospheric transport inversions assimilation techniques are often employed to invert the transport. Another example just follows on page 2, line 1: the inversion method does not provide information on the emissions; it is rather the observations that provide information. I suggest the authors (some of the co-authors are native or close-to-native speakers and some are world-leading experts in inverse modelling) carefully go through the manuscript to improve the language and correct these misunderstandings of how transport inversions work.*

We apologize for the misunderstandings and inconsistencies that arose as a consequence of the weak formulation that existed in the manuscript. In this revision, we have phrased our text more carefully, and also had the full paper language edited by a native English speaker. Moreover, we tried to make our descriptions more clear using new labeling.

*The main problem of this manuscript, however, is its aim, as the authors state: 'The aim is to introduce the set-up and choices for an inversion system, which will be used in long term studies and presented in an accompanying paper.' Is this really sufficient for a publication in its own or could that have been merged into the accompanying paper that looks at long-term methane emissions and trends. And how much do we really learn from a sensitivity study covering a very short time period (5 months) with no interannual nor seasonal variability for an inversion aiming to analyse long-term emission fields and trends? On top of that the CTE-CH$_4$ system has essentially already been published elsewhere (see Tsuruta et al., 2015).*

After reflecting on the comments from the reviewer, and following the suggestions, this paper was merged with the accompanying paper. Although experiments were carried only for a short time period, the findings were important and could hold also for experiments on longer time periods. Since the previously published Tsuruta *et al.* (2015), the system was further developed, and this paper extends the analysis with the new version of the model.

*- Why did the authors finally choose S1 and S5 for the long-term experiment in the accompanying paper? Based on the evaluation in this manuscript here there are not any indications to rule out any of the tested configurations. At least the authors do not provide any objective reasons.*

We agree with the reviewer that the choice of the simulations to be extended over the longer time period was based on expert judgment, and not so much on derived metrics. In the end, we did not find specific reasons to choose other set-ups based on uncertainty estimates and agreement with the NOAA in situ observations. Therefore, in the new manuscript, we decided to present each estimate as an equal realization of the surface fluxes. We hope the reviewer agrees that this is a more balanced representation of our results.

*- Is there anything the inverse modelling community can learn from the experiments here in general? Maybe the results of some of the experiments can be analysed in more depth such that the findings can be generalised?*

We acknowledge that some of the experiments were specifically meant to test the configuration of this system, following a custom of designing and presenting a new inverse modelling framework. In addition to the real flux estimates that are interesting in themselves, we also believe that the sensitivity to vertical transport, and the attempt to separately estimate biosphere and anthropogenic fluxes are useful to other, similar attempts, given that the sparsity of systems that perform inverse modeling of CH$_4$ fluxes. In the revised manuscript, sensitivities to those were analysed in depth based on multi-year simulations and validation with model independent observations. Moreover, the extensive evaluation presented here sets a target for future studies that we also find important.

*- It would be illustrative to run some longer experiments to account for seasonal and interannual variability for some selected set-ups.*

> This is an excellent suggestion, and we agree that our system would profit from further tests of robustness and sensitivity. As we mention in the revised manuscript, testing different prior emission patterns as well as varying the OH-sinks over time would be the first things to try. However, to benchmark our system and describe it for future reference, we feel that the revised manuscript already is quite extensive, which is merged with the accompanying paper. In the revised paper we present results from longer-term experiments to examine the robustness and sensitivity of the estimates on seasonal and interannual variabilities to the number of parameters, the vertical mixing schemes in the transport model, and briefly to the set of observations. We plan to continue using our system at FMI for the near future, and we hope the reviewer allows us to undertake these follow-up tests in a future study as well.

*- Although the title suggests that the inversion system focuses on Europe the analysis of the results in the manuscript does not. What is the effect of the zoom over Europe? And although you zoom over Europe you only distinguish between four regions.*

> The reviewer correctly remarks that our paper presents results for the full globe, rather than just focusing on Europe. We applied the zoom over Europe, since the observation network in Europe is most dense and we have a primary interest to study European $CH_4$ fluxes with our system in the future. With this higher resolution, $CH_4$ abundance at closely located sites can be resolved separately, taking into account meteorological parameters and transport of each grid cell. However, the regional inversion over Europe is only trustworthy if we can realistically constrain the inflow and outflow of $CH_4$ across regional boundaries, and for that we need to take into account the fluxes in the rest of the world. The current paper thus lies the foundation for future regional inversions. Note that the identifier "Europe" in our title refers to the origin of the CarbonTracker branch we use (to distinguish it from NOAA's CarbonTracker) and does not necessarily identify the focus of the study.

*- In Fig 4 plotting the relative differences in the uncertainty estimates would be much more illustrative.*

> We thank the reviewer for this suggestion. The figure was modified accordingly, and moved to supplementary material of the accompanying paper.

> **See Supplementary Material of accompanying paper, Fig. S3.**

*- In Fig 5 the fit against the observations when transporting the posterior fluxes is not very impressive in this kind of plot. Maybe time series for some selected stations would illustrate the improvement much better.*

> Here we aimed to illustrate the large scale and seasonal patters, and less focus on the agreement at individual stations. The format also corresponds to the presentation of residuals in other CarbonTracker applications, where the format is generally well appreciated. The requested time series are now part of the revised Figure 2, although we decided to retain the look.

---

## Author Comment (AC2) · 13 Dec 2016

Authors' response to anonymous referee #2

In the following, referee's comments are in *italic*, authors' responses in normal font, and references (page, line, figure, and table number) to the revised manuscript in **bold**. Please note that this paper was merged with the accompanying paper, following the referees' comments and with approval from the Topical Editor. The summary of this paper was included in the Supplementary Material of the accompanying paper.

*Although such sensitivity tests are undoubtedly important, the authors need to be clearer about how the outcomes of this work are of benefit to the wider inverse modelling community. In order to make the work more generally applicable perhaps the authors could provide some comparison of the relative importance of the input parameters, through a global sensitivity analysis for example. As it stands, the paper attempts to provide a justification for a particular model set-up to be used in the companion paper, but I wonder whether this is enough to justify a paper of its own, or whether this information should rather be included as a supplement to the companion paper.*

*Overall, I found the manuscript to be a little vague on what the outcomes of the sensitivity tests are, with a focus on qualitative rather than quantitative discussion. The manuscript was let down slightly by a number of grammatical errors or poorly constructed sentences, which may be why the key messages of the work are not clearly conveyed.*

We acknowledge that some of the experiments were specifically meant to test the configuration of this system, following a custom of designing and presenting a new inverse modelling framework. Following the excellent suggestions from the reviewer, we decided to merge this paper with the accompanying paper. We hope the reviewer agrees that the sensitivities to vertical transport, and the attempt to separately estimate biosphere and anthropogenic fluxes, presented in the revised manuscript, are sufficiently discussed and useful to others, given that the sparsity of systems that perform inverse modelling of $CH_4$ fluxes.

We apologize for the inconsistencies that arose as a consequence of the weak formulation that existed in the manuscript. In this revision, we tried to more carefully phrase our text, and have also had the full paper language edited by a native English speaker. Moreover, we tried to make our descriptions more clear using new labelling.

General comments:
*The paper focuses on sensitivity tests of various model inputs and parameters, and selects two models as a consequence of these tests. However, all tests assume the same model-data mismatches (mdm), which raises two major issues:*

*1. Clearly the mdm values are another input to the inversion which will change the form of R and thus the cost-function minimization. The impact of the mdm values on posterior emissions has been examined many times before (e.g. Michalak et al., 2005; Trudinger et al., 2007), with the studies commenting on the importance of these error terms. It seems a little odd therefore that this crucial component of the inversion is ignored in the sensitivity tests, given the somewhat arbitrary nature of their assignment. Furthermore, the observation error correlation structure would also impact on the solution, but I was unable to find any discussion of this in the manuscript.*

This is an excellent comment and suggestion. In this study, we chose mdm based on quality of the observations, site types (mbl, land, tower, etc), and transport model error from forward runs, and a previous study by Bruhwiler *et al.* (2014), which used a similar system. Although the choice of the mdm values was somewhat arbitrary, they were generally targeted to have posterior Chi-squred values (Michalak *et al.*, 2005) spread around 1 (see also response to second referee of the accompanying paper on Point 3 of Scientific concerns). From the Chi-squared statistics, we found that the chosen mdm values were within the expectations to some extent. We will continuously develop the method to choose mdm values, but we hope the reviewer accepts the current choices.

For observation error correlation, we assumed all observations are independent of each other. However, it is known that observations are spatially and temporally correlated to some extent. The spatial correlation could be accounted for by considering e.g. distance between the sites. To take temporal correlation into account, further development is needed on the propagation of the observation covariance matrix. In the revised manuscript, we elaborated these issues further.

**See e.g. Pg. 11 line 9-11 of the accompanying paper.**

*2. The decision to select the 2 chosen models S1 and S5 appears to be dependent on the posterior mismatch to the*

*observations. However, given this posterior is itself dependent on the chosen mdm, it is conceivable that under a different set of assumptions one would select a different model instead of S1 or S5. Since the values of mdm appear to be entirely down to investigator choice, I cannot see how the paper can propose an "optimal" inversion system that would be applicable beyond the specific case examined here.*

*In the introduction it is stated that the aim of the paper is to "introduce the set-up. . .for an optimally working methane inversion system." However, I am not convinced that this is achievable from only seven different inversion configurations. Comparatively, one configuration may be better than the other six, but it would require a much more in-depth analysis to find the "optimum" configuration. For instance, some combination of configurations S2, S4 and S7 could provide a better match to the observations. However, performing the sensitivity analysis in the localised way of this work means that such a conclusion cannot be reached.*

*It may be that the choice of S1 and S5 is justified but any clear evidence to support this conclusion was either lost in the text or not present. In fact, Figure 7 would appear to suggest that there is very little to distinguish between the majority of configurations at both the global and continental scale.*

> We agree with the reviewer that the choice of the set-up was based on expert knowledge rather than quantitative analysis. Also, the chosen set-up would not be optimal for the system. To find an optimal system, we need a much more extensive sensitivity analysis, including that of mdm, although we may not find an optimal set-up even after that. In this study, we did not find specific reasons to choose other set-ups than S1 and S5 based on uncertainty estimates and agreement with the NOAA in situ observations. Therefore, in the new manuscript, we decided to present each estimate as an equal realization of the surface fluxes. We hope the reviewer agrees that this is a more balanced representation of our results.

*Specific comments:*

*Page 7, Lines 5-7: What was this "information provided by the experimentalists"? How many continuous day or night time observations were assimilated per site per day? What was the form of the observation error covariance matrix (e.g. diagonal?) If it is diagonal, is this assumption justified? In general some key details appear to be missing.*

> For the selection of background observations, observation flags (measurement quality and assessment of background) provided by contributors were used. For example, for NOAA observations, observations without obvious problems during collection or analysis were chosen. Daily means from the selected observations were used in the system, and therefore, the number of observations per day was one.

> For the observations error covariance matrix, we did not assume any correlation between the observations, i.e. the matrix was diagonal. However, it is likely that some observations are temporally and spatially correlated. (see also above response to General comments 1).

> **Additional information was added in the accompanying paper. See Pg 8, line 14-15, 27-29.**

*Page 7, Lines 14-16: Given the TM5 model is run at higher resolution over Europe, one might assume this would lead to a reduced representation error for European stations, i.e. 1x1 degree boxes might be able to represent a point in space better than a 4x6 degree. However, the mdm values appear to be the same whatever the model resolution at each site. Is there a reason for this?*

> This is a very interesting point. We agree with the reviewer that resolving at higher transport model resolution generally reduces transport model error as it resolves the meteorological parameters and atmospheric mixing better, and also reduces "spreading" of emissions over large boxes (smearing). However, sites in Europe are also notoriously difficult to model due to their locations and uncertainty in emission sources. Furthermore, mdm also includes natural variability in measurements. Sites with small transport model error may have large mdm, if measurement variability is high. Therefore, we find that the mdm cannot be defined only based on the transport model resolution, and left the mdm of European sites not exceptionally small.

*Page 7, Line 17: ". . .for the sites that appeared problematic in the inversions. . ." What meant by "appeared problematic"? Tuning the mdm values post-inversion is surely unacceptable as a violation of Bayes rule. What is the justification for 75 ppb? If the data is problematic why not just discard it completely?*

> We agree with the reviewer that the mdm values should be chosen before inversion. In this study, the mdm was defined before inversion mainly based on quality of the observations, site types (mbl, land, tower, etc), and

transport model error from forward runs, and from a previous study by Bruhwiler *et al.* (2014), which used a similar system. From those, we learned which sites the model can, or can not, properly represent.

We acknowledge the concern of the reviewer about the effects of those sites with high mdm. It is indeed questionable how much information those observations provide to constrain the emissions. However, we did not simply remove them because we believed that some information could be useful. For example, for regions where observation network is sparse, the observation network is sparse, our emission estimates would act as a compensating effect by removing those observations, which may or may not be supported by the observations. Although we did not test thoroughly the effect of those observations by e.g. removing them, we hope the reviewer approves the use of these observations in the study.

*Page 14, Lines 1-3: "S3 posterior mole fractions matched the observations best. . .in other words the additional information from the continuous observations was useful in gaining better agreement with NOAA observations." But according to Table 2, S3 is the configuration with discrete observations only, so the above statement cannot be right. It doesn't seem very surprising that S3 matches the NOAA observations better when these are the observations that have been used to constrain the emissions. Surely it would be more helpful to compare to an independent dataset rather than those that have already been used to derive the emissions.*

*Page 14, Lines 24-25: "Indeed the additional observation uncertainty increased the emissions uncertainty also." I fail to see how increasing the number of data points (however uncertain) would increase uncertainty, and this is not backed up by Table 4 which shows the inversion with only discrete observations (S3) has the highest emissions uncertainty.*

Response to the above two comments on page 14:

We apologize for the misunderstanding and confusion that arose by poor phrasing. The paragraph was revised and included in the Supplementary Material of the merged paper as follows:

Removal of continuous observations decreased mean posterior anthropogenic emissions by about 70% in temperate North America and in southwest and east Europe. The decrease was partially compensated by an increase in biospheric emissions; for the North American temperate region, posterior biospheric emissions were about 100% larger without assimilating continuous observations, and the estimates were similar to the prior. Furthermore, the decrease was also compensated by >50% increase Asian tropic emission estimates. However, differences in biospheric emissions in the Asian temperate region were small. The reason could be that the discrete observations may have had little effect on the biospheric emissions, as the observations were located near anthropogenic sources. Therefore, the inversion less sensitive to biospheric emissions when continuous measurements are not assimilated. The effect of removing continuous observations was also significant in the uncertainty estimates, which were larger for anthropogenic emissions than for biospheric emissions. The posterior uncertainty for global anthropogenic emissions was about two times larger in the inversion not assimilating continuous observations, and the largest differences were found in the North American temperate and Asian temperate regions, and in southwest Europe. The posterior biospheric emission uncertainty was about three times larger in North American boreal, about twice as large in Asian temperate, and about 20% larger in North American temperate, Eurasian boreal, and Asian tropical regions than the estimates using continuous observations. These results indicate that improving prior estimates is important, especially for regions where observations are sparse.

*Page 16, Line 7-8: "Thus, improving the prior estimates is important even when using an inverse model in the absence of observations." I assume this is a case of poor phrasing, but I am intrigued as to how one would perform inverse modelling in the absence of observations.*

The sentence meant that, improving prior estimates is important especially for regions where the observation network is sparse. The phrase is revised and included in the Supplementary Material of the new manuscript.

**Text revised: Pg . 5, line 17-18 of  Supplementary Material of the accompanying paper.**

*Figure 2: I appreciate the Asian tropical plot is supposed to show the large variability with an ensemble size of 20, but is there a way of conveying the same message without it looking quite so messy? It might make the plot a little easier to interpret.*

Here, our focus was to illustrate the variability of weekly estimates between random draws. Simply showing

e.g. a range of estimates could mislead how estimates vary between weeks. To better illustrate the differences between E20 and E500, the estimates of E500 were plotted in thicker lines, although the we decided to retain the look.

**See Fig S1 in Supplementary material of the accompanying paper.**

---

## Author Comment (AC3) · 13 Dec 2016

This paper is merged with the accompanying paper, following the referees' comments and with approval from the Topical Editor, and we titled the new manuscript as "Global methane emission estimates for 2000-2012 from CarbonTracker Europe-CH$_4$ v1.0".